# STEERING DIFFUSION TRANSFORMERS WITH SPARSE AUTOENCODERS

## ABSTRACT

Diffusion transformers (DiTs) now underpin state-of-the-art image generation, yet their internal mechanisms remain less understood than those of large language models. Sparse autoencoders (SAEs) offer a route to both mechanistic interpretability and controllable editing, but prior SAE work has focused largely on U-Net–based diffusion models. We present, to our knowledge, the first systematic study of SAE-based feature steering in DiTs and show that aligning interventions with the dynamics of the residual stream reveals and exploits their causal structure. Our approach proceeds in three steps. We first develop multi-layer steering that operates across a feature's natural persistence interval, strengthening edits while suppressing artifacts. We then introduce a simple similarity-based presence test to detect when and where a feature is active in the residual stream, which guides layer selection for effective interventions. Finally, we train SAEs on transformer-block updates and residual states and use the resulting features with our steering strategy to map which components of the DiT exert the strongest downstream influence on targeted concept edits. Together, these advances clarify how DiTs write, propagate, and read semantic features, and they provide a practical recipe for high-fidelity feature-level editing in generation.

## 1 INTRODUCTION

The quest for model interpretability has led to significant advances in understanding the internal mechanisms of deep neural networks. Among the most promising techniques are sparse autoencoders (SAEs) (Bricken et al., 2023; Cunningham et al., 2023), which have been successfully employed to uncover disentangled and human-understandable features within large language models (LLMs) (Arad et al., 2025). While transformative in the language domain, the application of SAEs to vision models, particularly the increasingly complex text-to-image generators, remains a nascent and challenging field. Early works have started exploring the intermediate representations of the visual stream (Surkov et al., 2025a; Cywiński & Deja, 2025) itself during generation and show that steering these features enables reliable control over generation in U-Net–based diffusion models.

However, the state-of-the-art in text-to-image generation has shifted to diffusion transformers (DiTs) (Peebles & Xie, 2023; Esser et al., 2024). Unlike U-Nets (Rombach et al., 2022; Podell et al., 2023), which exhibit a natural down–middle–up block structure, DiTs consist of long sequences of transformer layers with no obvious division of labor. This raises a fundamental question: *where in the residual stream are high-level visual features written, and how can they be causally steered?* Initial attempts to train SAEs on DiTs show low reconstruction error, but steering those features with the single-layer interventions used in LLMs or U-Nets often fails, producing weak or inconsistent edits Surkov et al. (2025b). Without effective steering, it becomes impossible to validate the causality of features, leaving interpretability in DiTs an open problem.

In this work, we argue that these failures do not reflect a limitation of SAEs, but rather a mismatch between the steering procedure and the residual stream dynamics of transformers. In DiTs, features are progressively written, read, and rewritten across many layers. Injecting a feature only once, at its "discovery" layer, does not guarantee persistence in the residual stream for downstream readers. This motivates the need for a residual-stream–aware steering methodology.

**Contributions.** We introduce a framework for steering and interpreting diffusion transformers:

1. **Residual Stream API.** We develop a similarity-based method to track when and where features are present in the residual stream, enabling us to distinguish writer from reader layers and to steer more effectively.
2. **Multi-steering.** We propose to steer not at a single layer, but across the interval where a feature naturally persists, ensuring robustness and preventing vanishing. This unlocks the causal power of SAE features in DiTs.
3. **Experiments on RIEBench.** We evaluate our steering methods on RIEBench, a recent representation-based editing benchmarks for text-to-image diffusion models (Surkov et al., 2025a) and thereby show that specific blocks (e.g., Blocks 3–4 in Flux) write high-level semantic features, while others play little causal role. This highlights the importance of being able to steer for interpretability.

By combining feature discovery with residual-stream–aware steering, we establish the first reliable framework for causal interpretability in diffusion transformers. This opens the door to causal analysis of DiT features through vector-based interventions and is agnostic to the particular method used for feature discovery.

*We conduct all our experiments on FLUX-Schnell and SD3.5-Large-Turbo, which are few-step distilled (Sauer et al., 2024) versions of FLUX (BlackForestLabs, 2024) and SD3 (Esser et al., 2024).*

## 2 RELATED WORK

**Sparse autoencoders.** Sparse dictionary learning methods have recently gained traction for identifying interpretable directions in deep neural networks, including large language models (LLMs) (Bricken et al., 2023; Cunningham et al., 2023) and vision–language models (VLMs) (Joseph et al., 2025). Several SAE variants have been proposed, such as TopK (Gao et al., 2024), Batch-TopK (Bussmann et al., 2024), JumpReLU (Rajamanoharan et al., 2024), and Matryoshka (Bussmann et al., 2025). In LLMs, sparse dictionary learning has been used to uncover interpretable circuits and characterize how different layers interact (Marks et al., 2024). For LLMs Arad et al. (2025) show the importance of selecting the right SAE features for steering.

In this work, we train multiple SAEs on sequential layers of a DiT, revealing their functional roles. This provides a basis for further research on the interplay between DiT components.

**Interpretability of diffusion models.** Surkov et al. (Surkov et al., 2025a) apply SAEs to a distilled SDXL-Turbo model, identifying interpretable features, enabling causal interventions, and showing that one-step features can generalize to multi-step settings. Cywiński & Deja (2025) demonstrate how SAE features can be used for diffusion unlearning. Kim et al. (2024) extend this approach by training timestep-specific SAEs on unconditional samples, linking learned features to class-level information. Kim & Ghadiyaram (2025) show that SAEs trained on the text embeddings used for conditioning are effective for steering based image manipulation. Daujotas (2024) shows that SAEs trained on CLIP image embeddings also are effective for steering based image manipulation for models like (Razzhigaev et al., 2023) that can be conditioned on image embeddings.

However, these works either primarily target U-Nets and lack extensive evaluation or only decompose conditioning inputs to the diffusion models. Our work addresses this gap by applying SAEs to DiT components and systematically evaluating the causal effect of the extracted features.

**Editable image generation.** Diffusion-based image editing has become an active research area. Approaches such as (Wu et al., 2024; Couairon et al., 2022; Zhu et al., 2025) employ prompt conditioning and masking, while others such as Gandikota et al. (2024) train LoRA adapters, called concept sliders, to modify generated images. In their follow-up work, Gandikota et al. (2025) present a way to train concept sliders without human supervision by leveraging the principal components of images generated from a single prompt but with changing input noise. Principal components are computed by embedding the images using an embedding model like CLIP (Radford et al., 2021).

In contrast, our work focuses on interpretable steering, which enables fine-grained and precise editing that is not achievable through prompting and does not require retraining for every new edit.

## 3 BACKGROUND

**Sparse autoencoders.** We leverage sparse autoencoders to map internal representation vectors into a sparse latent space.

Following the notation of (Surkov et al., 2025a), the encoder is defined as:

$$\text{ENC}(h) = \sigma\big(W^{\text{ENC}}(h - b_{\text{pre}}) + b_{\text{act}}\big),$$

where $h \in \mathbb{R}^d$ is the input vector to be decomposed, $\sigma(\cdot)$ is an activation function, $b_{\text{pre}} \in \mathbb{R}^d$ and $b_{\text{act}} \in \mathbb{R}^{n_f}$ are learnable bias vectors, and $W^{\text{ENC}} \in \mathbb{R}^{n_f \times d}$ is a learnable encoder matrix. The dimensionality $d$ corresponds to the size of the input representation, and $n_f$ denotes the number of features in the sparse latent space.

The decoder is defined as

$$\text{DEC}(s) = W^{\text{DEC}}s + b_{\text{pre}},$$

where $s \in \mathbb{R}^{n_f}$ is a vector in the sparse latent space (typically the output of ENC), and $W^{\text{DEC}} \in \mathbb{R}^{d \times n_f}$ is a learnable decoder matrix.

The encoder-decoder pair is trained to minimize a reconstruction loss subject to a sparsity constraint. In this work, we use the Top-$k$ activation function, which preserves only the $k$ largest values of the input vector while setting all others to zero. Gao et al. (2024) has shown that Top-$k$ activations prevent activation shrinkage, leading to more faithful and stable sparse representations.

**Diffusion transformers.** Diffusion transformers (DiTs) are a class of generative models that combine the diffusion framework with transformer backbones (Peebles & Xie, 2023; Esser et al., 2024). Unlike U-Net–based denoising networks (Rombach et al., 2022; Podell et al., 2023), which rely on convolutional blocks and skip connections, DiTs use a pure transformer architecture. This design proved to scale better and be more effective for long-range dependencies modeling. Therefore it has been widely used for image and video generation.

Most modern DiTs, such as FLUX (BlackForestLabs, 2024) and Stable Diffusion 3 (SD3) (Esser et al., 2024) that we study in this paper, are trained using the rectified flow–matching objective by parameterizing a time-dependent velocity field that transports data to pure noise using a DiT:

$$\frac{dx_t}{dt} = v_\theta(x_t, t, \mathbf{c}), \tag{1}$$

where $t \in [0, 1]$ denotes the timestep, $x_0 \sim p_{\text{data}}$ the clean image, $x_1 = \epsilon \sim \mathcal{N}(0, I)$ noise, $\mathbf{c}$ denotes additional conditioning signals such as text prompts, and $\theta$ the trainable parameters of the DiT. The function $v_\theta$ is parameterized by a stack of transformer blocks that jointly process the noisy latent representation and the conditioning input, enabling information exchange across modalities. For more details on the training and inference of flow-matching DiTs consider Appendix A.

**DiT layers.** In addition to the timestep $t$, $v_\theta$ takes as input $x_t \in \mathbb{R}^{w \times h \times c}$ an interpolation between noise $\epsilon$ and latent-image $x_0$ (with $h = w = 128$ and $c = 16$ in FLUX) [1], and a textual prompt $\mathbf{c}$. The noisy latent $x_t$ gets converted into a sequence of visual tokens by patching it into $1 \times 1 \times 16$ patches and linearly up-projecting them to vectors $z_1, \ldots, z_{n_z} \in \mathbb{R}^d$, in which $n_z = w \cdot h$ denotes the sequence length and $d >> 16$ the latent dimension used in the model. Similarly, the textual prompt gets converted into a sequence of token embeddings $\mathbf{c}_1, \ldots, \mathbf{c}_{n_c} \in \mathbb{R}^d$, where $n_c$ is the number of text tokens, using both the CLIP text and T5 (Raffel et al., 2020) followed by a projection.

DiT layers update both the visual stream $z_1, \ldots, z_{n_z}$ and textual stream $\mathbf{c}_1, \ldots, \mathbf{c}_{n_c}$ using a self-attention operation, followed by a MLP with normalization layers in-between. Let $z^\ell$ denote the visual stream before layer $\ell$, $\mathbf{c}^\ell$ the textual stream before layer $\ell$ and $h^\ell = z^\ell \cdot \mathbf{c}^\ell$ their concatenation $z_1^\ell, \ldots, z_{n_z}^\ell, \mathbf{c}_1^\ell, \ldots, \mathbf{c}_{n_c}^\ell$. Then, DiT layers take the simple form $h^{\ell+1} = h^\ell + f^\ell(h^\ell)$, in which $f^\ell$ denotes the DiT layer. Spatial information inside of the visual stream and positional information in the textual stream are maintained using positional encodings. In this work, we train our SAEs on the updates performed by the multiple layers. E.g., layers 1–6 that is $f^1(h^1) + \cdots + f^6(h^6)$.

---

[1]To be precise, we are working with DiTs that are operating inside of the latent space of a variational autoencoder (Kingma & Welling, 2013), which is a computational trick enabling the synthesis of high-resolution images (Rombach et al., 2022).

**Distilled DiTs.** In this work, we mainly work with distilled DiTs (Sauer et al., 2024; Yin et al., 2024b;a), which are initialized from multi-step models (typically about 20–50 steps) but are trained to require fewer denoising steps (1–4). Surkov et al. (2025a) have shown that features learned in the distilled model transfer to the base model without additional training and that this thus is an effective approach for diffusion-/flow-matching model interpretability. We observe the same behavior.

*In the remainder of this paper we refer to the distilled versions FLUX-Schnell and SD3.5-Large-Turbo simply as FLUX and SD3.*

## 4 METHODS

In this section, we present our core methodological contributions for steering DiTs using SAE-derived features. We describe the full workflow, which begins with a trained SAE and leads to targeted steering interventions. Our approach is built on three main contributions:

1. **SAE feature selection**: a criterion for automatically selecting and combining the SAE features that are statistically relevant for a target concept.
2. **Residual-stream similarity analysis**: a technique to detect where features persist across the residual stream, which guides the choice of intervention layers.
3. **Multi-steering**: a novel intervention strategy that repeatedly injects features across multiple layers, ensuring they remain on the residual stream for all transformer blocks that may read them.

### 4.1 EXTRACTING INTERPRETABLE FEATURES FROM SAEs

In our steering setup, we assume access to a feature vector $v_c$ that, when injected into the model's activations, reliably introduces a target concept $c$. In practice, however, SAEs produce thousands of features, only a small subset of which are relevant for any given concept. This raises a key methodological question: **how can we automatically identify the SAE features most strongly associated with a target concept?**

We address this by introducing a correlation-based feature selection criterion similar to (Cywiński & Deja, 2025), which ranks SAE features by their statistical dependence on the presence of a concept. Importantly, correlation does not imply causation: the selected features are candidates whose causal effect must be validated through a steering procedure.

**Concept relevance score.** Formally, let $X \in \mathbb{R}^{W \times H \times C}$ be a random variable representing the input data (e.g., an image with width $W$, height $H$ and $C$ channels), $C \in \{0, 1\}$ a binary random variable indicating whether concept $c$ is present in the input and assume we have access to a SAE decoder matrix $W^{\text{DEC}} \in \mathbb{R}^{d \times n_f}$ whose columns (denoted by $v_i = W^{\text{DEC}}_{[:,i]}$) corresponds to learned features.

Let $S(i, X) \in [0, 1]$ be a deterministic function representing the normalized activation importance of the SAE feature $v_i$. Our key hypothesis is that feature $i$ is associated with concept $c$ if its expected importance increases when $c$ is present:

$$\mathbb{E}_{X \sim p(X|c)}[S(i, X) \mid C = 1] > \mathbb{E}_{X \sim p(X)}[S(i, X)]. \tag{2}$$

A natural definition for the feature importance is, for a feature $v_i$ and a set of inputs $\mathcal{D}$, the average activation over all image patches in $\mathcal{D}$:

$$s(i, \mathcal{D}) = \frac{1}{|\mathcal{D}|} \sum_{x \in \mathcal{D}} \frac{\mu(i, x)}{\sum_{j=1}^{F} \mu(j, x)}, \tag{3}$$

where $\mu(i, x) = \frac{1}{P} \sum_{p=1}^{P} a_i(x, p)$ is the average activation of feature $i$ across all spatial patches.

Using these estimates, we define the **Concept Relevance Score**:

$$R(i, \mathcal{D}_c, \mathcal{D}) = s(i, \mathcal{D}_c) - s(i, \mathcal{D}). \tag{4}$$

A high positive $R(i, c)$ indicates that feature $i$ is more active in the presence of $c$, making it a strong candidate for steering. Therefore, $R$ might be used to rank all SAE features and select the top-$k$ for steering. Depending on the concept, either a single feature may suffice (if the SAE encodes $c$ as a

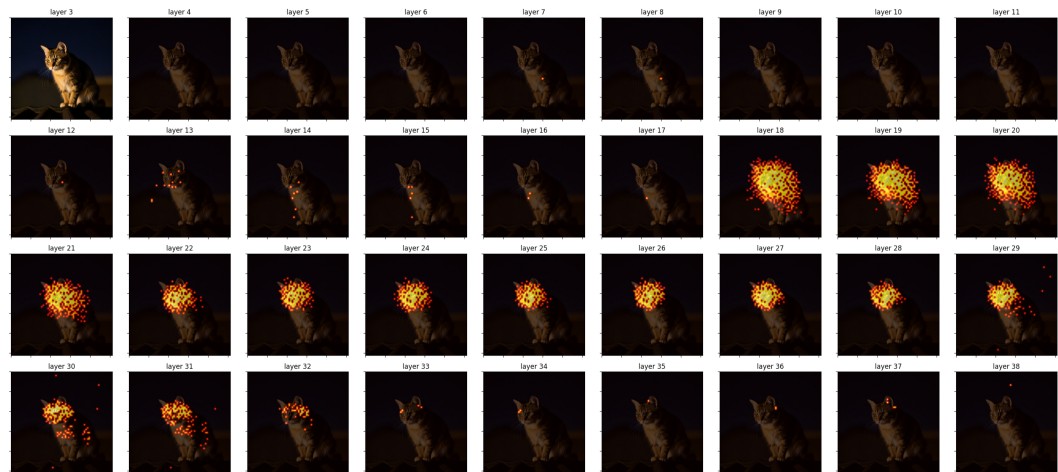

Figure 1: We compute cosine similarities between a cat feature learned by one of our SAEs with the residual stream at each spatial location at each layer. As a result with obtain the **feature lifespan (layer 18–32)** of the cat feature.

high-level direction), or multiple features may need to be combined (if $c$ is distributed across lower-level "building block" features). The number of selected features $k$ is therefore a hyperparameter.

To obtain a single steering vector, we compute a weighted linear combination of the selected features:

$$v_c = \sum_{i \in \text{Top-}k} \mu(i, \mathcal{D}_c) \cdot v_i, \tag{5}$$

where the weights correspond to the average activation of each feature in $\mathcal{D}_c$.

Finally, we perform **causal validation** by injecting $v_c$ into the model's activations and verifying whether it steers the generation toward the target concept (see Sec. 5).

### 4.2 RESIDUAL STREAM API

When DiT generate images, after a causal feature $v_c$ is written into the residual stream by some layer $l$, subsequent layers may read and reuse it to guide their computations. This raises a central question: *which layers rely on $v_c$, and for how long does it persist in the residual stream?*

Answering this requires a method to establish whether and where a feature is accessible throughout the network. We propose a simple but effective technique to analyze the persistence of features across layers. Given a target feature vector $v_c$, we compute its cosine similarity with the residual stream activations at all positions after each transformer block. A high similarity indicates that the latent representation contains a component aligned with $v_c$, meaning the feature is present in the residual stream.

Since SAE features are not orthogonal, cosine similarity values at locations where the feature is absent are not exactly zero. To address this, we establish an adaptive baseline: for each layer, we measure similarities on inputs where the target concept $c$ is absent, and we take the maximum observed similarity as a layer-specific threshold. A feature is then considered present at a given layer if its similarity exceeds this adaptive threshold. As Fig. 1 shows, this method produces clean heatmaps that reveal the trajectory of features through the network. For instance, for the feature "cat," we observe that it emerges at a specific "writer" layer $A$, remains detectable for several subsequent layers, and eventually vanishes around layer $B$. This provides evidence for a write–read dynamic: certain layers inject the feature, while others consume it before it dissipates.

**Implications for steering.** These insights directly suggest a multi-layer steering strategy. Instead of a single injection at the presumed writing layer, we plan to apply repeated interventions across the range of layers where the feature is naturally present. Beyond improving editing quality (see Sec. 5),

this approach establishes steering as a tool for interpretability, revealing when and where concepts are represented and propagated inside the DiT.

### 4.3 MULTI-STEERING

In diffusion transformers, the residual stream evolves from the initial noise input to the final denoised output. Each transformer block can *write* new features into the stream or *read* features written earlier. For a given concept feature vector $v_c$, some layers act as writers, while some subsequent layers act as readers that consume the information during denoising.

Steering seeks to introduce, edit, or suppress a target concept $c$ by manipulating its corresponding feature vector $v_c$. The common approach in prior work is to inject $v_c$ once into the residual stream at the layer $l$ where it was discovered. This simulates the effect of the writer and relies on downstream readers to make use of the injected feature.

We hypothesize that, because the residual stream is continuously updated, a feature written at layer $l$ can in principle be read at any subsequent layer. To ensure successful steering, the feature must therefore be maintained across the range of layers that consume it. This motivates our **multi-layer steering** approach, in which the intervention is applied not at a single layer but across a configurable interval of layers.

**Multi-steering.** Our multi-steering algorithm consists of modifying the standard generation process of a DiT by intervening in a selected interval $[A, B]$. Within this interval, the residual stream is augmented with the target feature vector $v_c$, scaled by a coefficient $\alpha$. The hyperparameters $A, B$ determine the range of intervention, while $\alpha$ controls the steering strength. For simplicity, we fix $\alpha$ across layers and timesteps, though adaptive variants are possible.

The procedure is summarized in Appendix B Algorithm 1. Note that if $v_c$ is extracted from layer $l$ and we choose $A = B = l$, then our algorithm reduces to the simple steering baseline used in prior work.

The common practice of single-layer steering, widely used in LLMs and U-Net–based diffusion models, proves suboptimal in DiTs, as it does not take full advantage of SAE-learned features. This distinction is crucial for interpretability: ineffective steering may give the false impression that a feature is low-quality or non-causal, when in reality the representation is valid but underutilized by the steering method.

## 5 EXPERIMENTS

Our experimentation aims to address two core research questions: i) Which blocks of a diffusion transformer contain interpretable features? ii) What is the most effective way to steer DiTs? We answer these questions using SAEs trained on DiT activations, applying our multi-layer residual stream steering methodology, and evaluating across multiple image editing tasks. We use Flux-Schnell and SD3-Large-Turbo, the distilled few-step versions of Flux-Dev and SD3-Large. Distillation improves efficiency while preserving feature generalizability, as shown in Surkov et al. (2025a).

### 5.1 STEERING UNLOCKS INTERPRETABILITY: FINDING CAUSAL BLOCKS

To achieve diffusion transformers interpretability, we need the ability to identify the computation performed by each layer, understanding which layers introduce high-level semantic features and how these features guide the image generation process. The image stream gradually evolves from noise to a final denoised image, with transformer blocks sequentially writing and reading features. We hypothesize that some blocks act as *writers* of semantic concepts (e.g., objects, materials, style), while subsequent blocks act as *readers*, propagating these features to the output.

To study this, we use targeted steering: we inject features learned by SAEs into selected blocks and observe the causal effect on the generated image. The magnitude of these effects identifies which blocks have the largest influence, enabling interpretability of the DiT's internal computation.

**SAE training.** Unlike U-Net models, which have a clear down-middle-up block structure, DiTs consist of dozens of sequential transformer layers. To balance granularity and interpretability, we

| Block | Change Object | | Add Object | | Change Content | | Change Color | | Change Material | | Change Background | | Change Style | |
|---|---|---|---|---|---|---|---|---|---|---|---|---|---|---|
| | 95% | max | 95% | max | 95% | max | 95% | max | 95% | max | 95% | max | 95% | max |
| Block 0 | 0.046 | 0.078 | 0.065 | 0.089 | 0.039 | 0.067 | **0.059** | **0.085** | 0.090 | 0.136 | 0.052 | 0.082 | 0.029 | 0.065 |
| Block 1 | 0.028 | 0.048 | 0.043 | 0.066 | 0.025 | 0.041 | 0.047 | 0.072 | 0.044 | 0.086 | 0.036 | 0.062 | 0.038 | 0.066 |
| Block 2 | 0.022 | 0.042 | 0.054 | 0.097 | 0.024 | 0.044 | 0.042 | 0.065 | 0.043 | 0.073 | 0.039 | 0.063 | 0.031 | 0.053 |
| Block 3 | **0.133** | **0.175** | **0.074** | **0.107** | 0.066 | **0.108** | 0.028 | **0.080** | 0.092 | **0.179** | **0.080** | **0.124** | 0.109 | 0.156 |
| Block 4 | 0.077 | 0.120 | 0.039 | 0.067 | **0.080** | 0.096 | **0.054** | **0.083** | **0.098** | 0.141 | 0.042 | 0.078 | **0.122** | **0.173** |
| Block 5 | 0.016 | 0.034 | 0.024 | 0.051 | 0.036 | 0.066 | 0.026 | 0.048 | 0.027 | 0.055 | 0.028 | 0.052 | 0.059 | 0.099 |
| Block 6 | 0.009 | 0.021 | 0.023 | 0.053 | 0.018 | 0.029 | 0.020 | 0.037 | 0.018 | 0.038 | 0.028 | 0.056 | 0.048 | 0.073 |
| Block 7 | 0.020 | 0.032 | 0.029 | 0.068 | 0.012 | 0.023 | 0.027 | 0.043 | 0.012 | 0.026 | 0.025 | 0.053 | 0.026 | 0.043 |
| Block 8 | 0.014 | 0.026 | 0.023 | 0.043 | 0.020 | 0.029 | 0.036 | 0.057 | 0.023 | 0.046 | 0.046 | 0.067 | 0.030 | 0.059 |
| Block 9 | 0.005 | 0.010 | 0.011 | 0.024 | 0.012 | 0.021 | 0.028 | 0.039 | 0.022 | 0.033 | 0.030 | 0.044 | 0.023 | 0.036 |

Table 1: Comparison of dinov2_result across blocks and tasks (values in absolute terms). Each task reports 95th percentile (shaded) and maximum.

divide each DiT into 10 blocks of consecutive layers. For example, in Flux (57 layers), each block contains 6 layers except the last, which has 3 layers; in SD3 (38 layers), each block contains 4 layers except the last, which has 2 layer. All subsequent analysis is conducted at the block level.

We train top-$K$ SAEs with expansion factor $f = 4$ and $K \in \{10, 20, 40, 80\}$. SAEs are trained on individual latents *block activation*, i.e. the delta added to the residual stream by the block iself, isolating features introduced by this block thus providing clearer attribution to layers as feature writers. SAEs are trained using l2-loss with an auxiliary reconstruction error term (fixing $auxk = 256$) on `laion-5b` prompts, using patch-level latents as inputs. We iterate until convergence, up to 1M prompts.

**RIEBench evaluation.** We evaluate steering performance on RIEBench Surkov et al. (2025a), which provides prompt pairs for diverse editing tasks (e.g., object addition, object replacement, attribute modification). For each task and prompt, we test multiple steering hyperparameters: $K \in \{10, 20, 40, 80\}$, which SAE to use, *block* which block to select, $n_f \in \{1, 3, 5, 10, 20, 40, 80\}$ num of SAE features selected, $\alpha \in \{0.25, 0.5, 1, 2, 5, 10\}$ steering coefficient, yielding 1680 configurations per prompt.

Each editing sample consists of an `original` prompt (e.g. *"A dog running in a field"*) and an `editing` prompt (e.g. *"A cat running in a field"*, corresponding to the `dog` $\rightarrow$ `cat` transformation). We will refer to the images generated with such prompts are the `original` and `edited` images. For each editing task, we:

1. Generate `original` and `edited` images in one step (to match our SAE training settings), use *GroundedSAM2* Ren et al. (2024) to get the segmentation mask of the subject of the editing and cache latents internal activations after all layers.
2. Apply the SAE to both `original` and `editing` latents, and use our feature selection criterion to get the top $N = 100$ features
3. Take the feature set difference `editing` minus `original` to keep only the top-$n_f$ features which contribute to the editing and not already present in original.
4. Steer `original` generation with residual-stream interventions (using 1), with $A, B$ set to the final layer of each block (will be referred to as *Simple* steering in section 5.2),
5. Repeat it to sweep over hyperparameters grid.

We evaluate a steering primarily with delta DinoV2 Oquab et al. (2023) similarity

$$\Delta(\text{dino}) = \text{dinov2\_sim}(\texttt{result}, \texttt{target}) - \text{dinov2\_sim}(\texttt{source}, \texttt{target}); \quad (6)$$

which corresponds to the semantic-visual similarity to the edited image, looking at its increment between the steered and the original images. Other metrics that we report in the supplementary material are delta *CLIP-similarity* Radford et al. (2021), to measure the alignment with the editing prompt, and *LPIPS* Zhang et al. (2018), a proxy for the perceptual distance between two images.

**Results.** Table 1 summarizes results across editing tasks. For Flux, Block 3 (layers 18–23) and Block 4 (layers 24–30) consistently yield the largest DINOv2 improvements across tasks. These blocks appear to concentrate semantic feature writing, confirming their causal role in guiding high-level content. We see a similar pattern for SD3.5-Large-Turbo (see Table 3 in the appendix), where

block 5 often emerges as the one leading to biggest increase in Dinov2-similarity, reinforcing the idea that central layers contribute to high-level semantic representations in DiT. We therefore conclude that such blocks serve as key writers for semantic features.

## 5.2 How should we steer our DiT?

After identifying blocks that are effective for introducing causal features, we now investigate which is the best performing steering strategy for DiTs. We evaluate the following strategies: a single injection at the last layer of the block (**Simple**, $A = B$ in algorithm 1), multi-steering at every layer starting from the last layer of the block (**All**, $A$ is the first layer of the block and $B$ is the last DiT layer), multi-steering at every layer within the block (**Block**, $A < B$, $A$ and $B$ are, respectively, the first and last layer of the block) and the residual stream-aware steering, which leverages our residual stream API to determine the layers for steering (**ResAPI**, see section 4.2).

Note that, as similarity threshold, we compute similarities on `original` generation and take max similarity for each layer as the minimum threshold to consider the feature activated. For Flux-schnell core block 3 the most commonly selected blocks compose the range $[18, 19, \ldots, 31]$ while for Stable-diffusion 3 they are $[20, 21, \ldots, 30]$.

For each edited prompt, we extract SAE features using our selection method, then apply each steering approach with its recommended set of layers. The steering hyperparameters are the same as in section 5.1, except for adding higher steering weights to cover a wider range up to $50.0$. For each method, we sweep steering hyperparameters and select the configuration that achieves the highest semantic similarity (measured with DINOv2) to the target prompt as the representative of best achievable performance by the steering strategy.

**Quantitative results.** Since edits vary in difficulty and baselines vary in strength, raw similarity scores are not directly comparable across prompts. To ensure a fair comparison, we perform a rank-based evaluation: for each prompt, steering methods are ranked according to delta DinoV2 similarity, then we compute the average across prompts to obtain a robust measure of comparative performance.

Table 2 reports average ranks across prompts for all steering methods. We report Dinov2-similarity, CLIP score and LPIPS in appendix to show the effective similarity increment, which indicate successful editing. Residual-aware and Block steering consistently outperform simple and all layer steering in all our metrics, while also maintaining competitive LPIPS scores. All-layer steering achieves often introduces global artifacts, reflected in worse LPIPS and simple steering stands as the less reliable one, with some failure showed in figure 2. We conclude that multi-layer strategies, combined with informed layer selection, boost steering perfromance and that single layer steering should not be deemed as representative of the SAEs learned features.

**Qualitative analysis.** Figure 2 illustrates typical outcomes. Naive steering often yields weak or barely visible edits, while residual-aware steering produces strong, localized edits (e.g. "dog → cat" or "red → blue car") without disrupting the background or other objects, confirming that our method not only improves performance but also enhances interpretability of how DiTs represent concepts. Refer to Appendix D Fig. 3 for more examples for various tasks.

## 6 Conclusion

Diffusion transformers are often deemed "hard to steer," obscuring the causal meaning of their internal features. Viewing them through the *Residual Stream API*, i.e., where features are *written*, persist across layers, and are later *read*, led us to a multi-layer intervention aligned with a feature's persistence interval. We show that SAE features in modern DiTs do carry causal leverage when steered at the right span: edits are stronger, more consistent, and less destructive. As a probe, this method maps where concepts are written and which layers exert downstream influence, clarifying the division of labor within DiTs. Practically, it enables precise feature-level editing for both generation and real-image manipulation.

By aligning interventions with where features truly live, we move DiT steering from ad hoc control to a robust, causality-oriented tool for scientific inquiry and controllable editing.

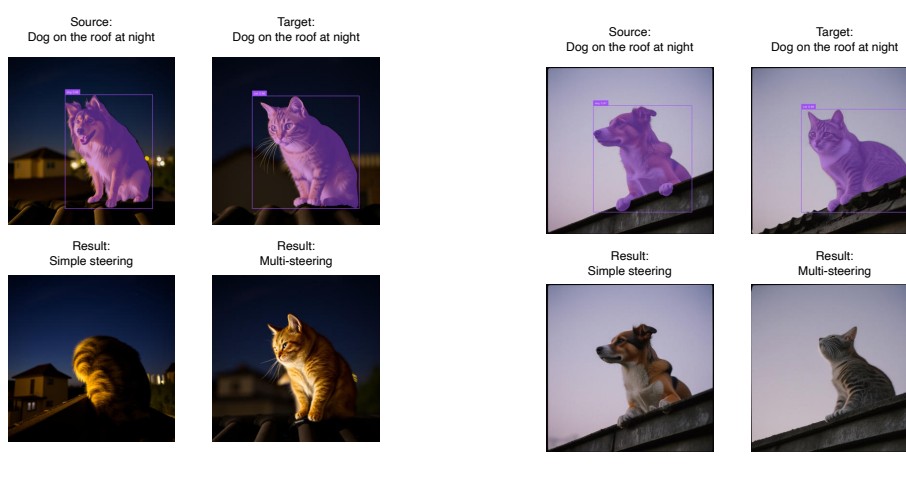

(a) Flux Schnell  (b) Stable Diffusion 3

Figure 2: RIEBench editing with Flux Schnell (a) and Stable Diffusion 3 (b). **Simple steering fails to reliably transfer the object, whereas multi-steering achieves the correct edit.**

| Model | Method | Change Object | Add Object | Change Content | Change Color | Change Material | Change Background | Change Style | Micro-Avg |
|-------|--------|--------|--------|--------|--------|--------|--------|--------|--------|
| Flux | Simple | 2.66 | 2.54 | 2.60 | 2.69 | 2.36 | 3.07 | 2.96 | 2.75 |
|      | ResAPI | **1.89** | 2.23 | **2.00** | **2.03** | **2.12** | 2.17 | **2.15** | **2.09** |
|      | Block | 2.17 | 2.35 | 2.65 | 2.71 | 2.58 | **1.80** | 2.48 | 2.35 |
|      | All | 3.26 | 2.84 | 2.75 | 2.52 | 2.88 | 2.88 | 2.38 | 2.77 |
| SD3 | Simple | 2.62 | 3.22 | 2.75 | 3.04 | 2.82 | **2.11** | 3.07 | 2.76 |
|     | ResAPI | 1.96 | **1.87** | **1.79** | 2.50 | **1.91** | 3.00 | **1.86** | 2.08 |
|     | Block | **1.67** | 2.09 | 1.95 | **1.78** | 2.00 | 2.17 | 2.12 | **2.00** |
|     | All | 3.57 | 2.70 | 3.40 | 2.48 | 3.04 | 2.47 | 2.90 | 2.73 |

Table 2: Average rank (lower is better) across steering methods and tasks, separated by model. Best per task in **bold**. Micro-Avg is weighted by the number of samples in each task, while Macro-Avg averages across tasks equally.

## ETHICS STATEMENT

This work advances the mechanistic interpretability for text-to-image diffusion transformers by training sparse autoencoders (SAEs) that discover latent features in an unsupervised way and enable feature-level generation control. Our aim is to advance transparent and trustworthy understanding of generative models. This research involves no human subjects, personally identifiable data, or confidential information.

**Unwanted content.** However, because SAEs discover features without supervision, a subset of discovered directions could correlate with sensitive concepts such as sexual or violent content. Similarly, unsupervised features inevitably reflect the statistical properties of their training corpus and may encode demographic or societal biases. Crucially, our primary experiments are conducted on FLUX and SD3, both diffusion models whose pre-training data was extensively filtered to exclude explicit or harmful imagery. As a result, they do not, by default, produce unwanted content, and our method does not weaken this property. We nonetheless recommend that downstream uses include careful feature curation, allow-lists, and human oversight. Further, we do not release features intended to target protected attributes and encourage independent bias and fairness evaluations prior to deployment.

## REPRODUCIBILITY STATEMENT

We build solely on publicly available, open-source, open-weight backbones (SD3 and FLUX), thus, our work is fully reproducible. Upon publication, we will provide implementations of our complete training and evaluation pipelines, including configuration files and exact hyperparameters. All of our SAEs are trained on intermediate states collected via forward passes on prompts of the guangyil/laion-coco-aesthetic-dataset on huggingface.

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

## A  DIFFUSION TRANSFORMER

Diffusion transformers (DiTs) are a class of generative models that combine the diffusion framework with transformer backbones (Peebles & Xie, 2023; Esser et al., 2024). Unlike U-Net–based denoising networks (Rombach et al., 2022; Podell et al., 2023), which rely on convolutional blocks and skip connections, DiTs use a pure transformer architecture. This design proved to scale better and be more effective for long-range dependencies modeling. Therefore it has been widely used for image and video generation.

**Training.**  Most modern DiTs, such as FLUX (BlackForestLabs, 2024) and Stable Diffusion 3 (SD3) (Esser et al., 2024) that we study in this paper, are trained using the rectified flow–matching objective. This is done by parameterizing a time-dependent velocity field that transports data to pure noise using a DiT:

$$\frac{dx_t}{dt} = v_\theta(x_t, t, c), \tag{7}$$

where $t \in [0, 1]$ denotes the timestep, $x_0 \sim p_{\text{data}}$ the clean image, $x_1 = \epsilon \sim \mathcal{N}(0, I)$ noise, $c$ denotes additional conditioning signals such as text prompts, and $\theta$ the trainable parameters of the DiT. The function $v_\theta$ is parameterized by a stack of transformer blocks that jointly process the noisy latent representation and the conditioning input, enabling information exchange across modalities.

Rectified flow models are trained to move along a straight-line probability path,

$$x_t = (1 - t)\, x_0 + t\, \epsilon, \tag{8}$$

with $x_0 \sim p_{\text{data}}$ and $\epsilon \sim \mathcal{N}(0, I)$. The target velocity is $x_0 - \epsilon$. The rectified flow–matching loss then becomes

$$\min_\theta \ \mathbb{E}_{t, x_0, \epsilon} \big[\, \|v_\theta(x_t, t, c) - (x_0 - \epsilon)\|_2^2 \,\big]. \tag{9}$$

**Inference.** During inference, we generate samples by numerically solving the ODE in reverse time using Euler's method (or higher-order solvers), starting from pure noise $x_1 = \epsilon \sim \mathcal{N}(0, I)$ and integrating backwards to obtain clean data $x_0$

$$x_{t_1} = x_{t_0} + (t_1 - t_0) \cdot v_\theta(x_{t_0}, t_0, c), \tag{10}$$

in which $t_0 > t_1 \in [0, 1]$.

**Distilled DiTs.**  In this work, we mainly work with distilled DiTs (Sauer et al., 2024; Yin et al., 2024b;a), which are initialized from multi-step models (typically about 20–50 steps) but are trained to require fewer denoising steps (1–4). Surkov et al. (2025a) have shown that features learned in the distilled model transfer to the base model without additional training and that this thus is an effective approach for diffusion-/flow-matching model interpretability.

**DiT layers.**  As defined above, in addition to the timestep $t$, flow matching DiTs take as input $x_t \in \mathbb{R}^{h \times w \times c}$ an interpolation between noise $\epsilon \in \mathbb{R}^{h \times w \times c}$ and latent-image $x_0 \in \mathbb{R}^{h \times w \times c}$ (with

| Block | Change Object | | Add Object | | Change Content | | Change Color | | Change Material | | Change Background | | Change Style | |
|---|---|---|---|---|---|---|---|---|---|---|---|---|---|---|
| | 95% | max | 95% | max | 95% | max | 95% | max | 95% | max | 95% | max | 95% | max |
| Block 0 | 0.041 | 0.065 | **0.063** | 0.091 | 0.039 | 0.075 | 0.040 | 0.080 | 0.060 | 0.105 | 0.055 | 0.071 | 0.014 | 0.032 |
| Block 1 | 0.032 | 0.070 | 0.053 | 0.102 | 0.029 | 0.061 | 0.032 | 0.101 | 0.049 | 0.093 | 0.052 | 0.079 | 0.012 | 0.025 |
| Block 2 | 0.033 | 0.045 | 0.035 | 0.074 | 0.030 | 0.056 | 0.026 | 0.064 | 0.015 | 0.033 | 0.039 | 0.053 | 0.030 | 0.068 |
| Block 3 | 0.021 | 0.053 | 0.037 | 0.075 | 0.021 | 0.037 | 0.031 | 0.076 | 0.028 | 0.068 | 0.052 | 0.079 | 0.054 | **0.109** |
| Block 4 | 0.039 | 0.065 | 0.030 | 0.056 | 0.041 | 0.065 | 0.032 | 0.098 | 0.023 | 0.058 | 0.034 | 0.052 | 0.046 | 0.074 |
| Block 5 | **0.060** | **0.114** | 0.056 | **0.102** | **0.044** | **0.090** | 0.033 | 0.066 | 0.035 | 0.068 | 0.045 | 0.077 | 0.055 | 0.073 |
| Block 6 | 0.023 | 0.058 | 0.026 | 0.051 | 0.033 | 0.054 | 0.038 | 0.102 | 0.019 | 0.046 | **0.056** | **0.090** | 0.038 | 0.071 |
| Block 7 | 0.029 | 0.041 | 0.033 | 0.052 | 0.033 | 0.054 | 0.035 | 0.092 | 0.011 | 0.034 | 0.051 | 0.078 | 0.054 | 0.078 |
| Block 8 | 0.015 | 0.035 | 0.025 | 0.042 | 0.025 | 0.043 | 0.038 | 0.095 | 0.016 | 0.040 | 0.042 | 0.064 | **0.066** | 0.085 |
| Block 9 | 0.022 | 0.043 | 0.050 | 0.069 | 0.004 | 0.033 | **0.170** | **0.187** | **0.186** | **0.190** | -0.054 | -0.018 | 0.063 | 0.085 |

Table 3: **SD3.5-Large-Turbo delta dinov2 similarity** between result and original images across 10 blocks and editing tasks. Each task reports 95th percentile and maximum values.

$h = w = 128$ and $c = 16$ in FLUX) [2], and a textual prompt $c$. The noisy latent $x_t$ gets converted into a sequence of visual tokens by patching it into $1 \times 1 \times 16$ patches and linearly up-projecting them to vectors $z_1, \ldots, z_{n_z} \in \mathbb{R}^d$, in which $n_z = w \cdot h$ denotes the sequence length and $d >> 16$ the latent dimension used in the model. Similarly, the textual prompt gets converted into a sequence of token embeddings $c_1, \ldots, c_{n_c} \in \mathbb{R}^d$, where $n_c$ is the number of text tokens, using both the CLIP text encoder as well as T5 (Raffel et al., 2020) followed by a linear projection layer.

DiT layers update both the visual stream $z_1, \ldots, z_{n_z}$ and textual stream $c_1, \ldots, c_{n_c}$ using a self-attention operation, followed by a MLP with normalization layers in-between. Let $z^\ell$ denote the visual stream before layer $\ell$, $c^\ell$ the textual stream before layer $\ell$ and $h^\ell = z^\ell \cdot c^\ell$ their concatenation $z^\ell_1, \ldots, z^\ell_{n_z}, c^\ell_1, \ldots, c^\ell_{n_c}$. Then, DiT layers take the simple form

$$h^{\ell+1} = h^\ell + f^\ell(h^\ell), \tag{11}$$

in which $f^\ell$ denotes the DiT layer. Spatial information inside of the visual stream and positional information in the textual stream are maintained using positional encodings. In this work, we train our SAEs on the updates performed by the layers, that is $f^\ell(h^\ell)$.

## B    MULTI-LAYER STEERING

We provide our multi-layer steering algorithm, Algorithm 1:

## C    STEERING COMPARISON: DINOV2 AND CLIP

| Model | Method | Change Object | Add Object | Change Content | Change Color | Change Material | Change Background | Change Style | Micro-Avg |
|---|---|---|---|---|---|---|---|---|---|
| Flux | Simple | 0.276 (0.392) | 0.123 (0.239) | 0.150 (0.391) | 0.104 (0.271) | 0.193 (0.249) | 0.143 (0.363) | 0.115 (0.360) | 0.158 |
| | ResAPI | **0.306 (0.396)** | **0.126 (0.231)** | 0.145 (0.348) | 0.118 (0.300) | 0.207 (0.289) | 0.175 (0.493) | 0.144 (0.476) | 0.198 |
| | Block | 0.292 (0.403) | 0.119 (0.295) | 0.136 (0.373) | 0.109 (0.318) | 0.196 (0.312) | 0.186 (0.507) | 0.131 (0.471) | 0.208 |
| | All | 0.205 (0.331) | 0.117 (0.237) | 0.127 (0.344) | 0.117 (0.255) | 0.178 (0.239) | 0.156 (0.498) | 0.137 (0.493) | 0.186 |
| SD3 | Simple | 0.120 (0.325) | 0.088 (0.168) | 0.132 (0.222) | 0.080 (0.133) | 0.069 (0.170) | 0.075 (0.256) | 0.071 (0.223) | 0.100 |
| | ResAPI | **0.198 (0.401)** | **0.122 (0.263)** | **0.183 (0.386)** | 0.100 (0.322) | **0.112 (0.284)** | 0.050 (0.247) | 0.114 (0.342) | 0.139 |
| | Block | 0.185 (0.431) | 0.119 (0.201) | 0.158 (0.357) | 0.147 (0.234) | 0.115 (0.260) | 0.081 (0.275) | 0.105 (0.330) | 0.160 |
| | All | 0.086 (0.262) | 0.090 (0.148) | 0.092 (0.183) | 0.115 (0.225) | 0.081 (0.211) | 0.070 (0.330) | 0.090 (0.313) | 0.114 |

Table 4: Steering comparison using Dinov2 (LPIPS) for Flux and SD3 models. Each cell shows dinov2 score with corresponding LPIPS in parentheses. Micro-Avg is computed across tasks weighted by sample count. Best per task is in **bold**.

## D    QUALITATIVE STEERING COMPARISON

---

[2]To be precise, we are working with DiTs that are operating inside of the latent space of a variational autoencoder (Kingma & Welling, 2013), which is a computational trick enabling the synthesis of high-resolution images (Rombach et al., 2022).

---

**Algorithm 1** Steering in a Diffusion Transformer

---

**Inputs:**
 $\epsilon_\theta$: Pre-trained Model
 $\{\text{TBlock}_l\}_{l=1}^L$: Layers
 $f_{\text{target}} \in \mathbb{R}^d$: Target vector
 $\alpha \in \mathbb{R}$: Coefficient
 $A, B$: Layer indices
 $\{t_i\}_{i=1}^T$: Timesteps
 $c$: Conditioning
**Output:**
 $x_0$: Final steered image

1: **procedure** STEERGENERATION$(z_T, \dots)$
2:  Let $z_{t_1} \sim \mathcal{N}(0, I)$
3:  **for** $i = 1, \dots, T-1$ **do**
4:   Let $t = t_i$.
5:   $h_0^{(t)} \leftarrow \text{Emb}(z_t, t, c)$
6:   **for** $l = 1, \dots, L$ **do**
7:    $h_{\text{out}} \leftarrow \text{TBlock}_l(h_{l-1}^{(t)})$
8:    **if** $A \leq l \leq B$ **then**
9:     $h_l^{(t)} \leftarrow h_{\text{out}} + \alpha f_{\text{target}}$
10:    **else**
11:     $h_l^{(t)} \leftarrow h_{\text{out}}$
12:    **end if**
13:   **end for**
14:   $\hat{\epsilon} \leftarrow \text{Proj}(h_L^{(t)})$
15:   $z_{t_{i+1}} \leftarrow \text{Step}(z_t, \hat{\epsilon}, t)$
16:  **end for**
17:  $x_0 \leftarrow \text{VAE\_Decode}(z_{t_T})$
18:  **return** $x_0$
19: **end procedure**

---

| Model | Method | Change Object | Add Object | Change Content | Change Color | Change Material | Change Background | Change Style | Micro-Avg |
|-------|--------|---------------|------------|----------------|--------------|-----------------|-------------------|--------------|-----------|
| Flux | Simple | 0.066 (0.395) | 0.020 (0.223) | 0.043 (0.386) | 0.026 (0.304) | 0.034 (0.300) | 0.041 (0.398) | 0.031 (0.353) | 0.038 |
| | ResAPI | 0.068 (0.440) | 0.022 (0.266) | 0.044 (0.397) | 0.032 (0.342) | 0.034 (0.345) | 0.050 (0.542) | 0.039 (0.510) | 0.050 |
| | Block | 0.068 (0.436) | 0.020 (0.279) | 0.042 (0.410) | 0.030 (0.341) | 0.036 (0.333) | 0.049 (0.533) | 0.035 (0.488) | 0.045 |
| | All | 0.058 (0.341) | 0.021 (0.273) | 0.044 (0.365) | 0.033 (0.328) | 0.034 (0.281) | 0.046 (0.556) | 0.041 (0.536) | 0.040 |
| SD3 | Simple | 0.039 (0.312) | 0.015 (0.153) | 0.026 (0.240) | 0.018 (0.141) | 0.019 (0.195) | 0.010 (0.274) | 0.015 (0.220) | 0.020 |
| | ResAPI | 0.052 (0.426) | 0.023 (0.244) | 0.037 (0.343) | 0.034 (0.398) | 0.028 (0.318) | 0.013 (0.181) | 0.021 (0.389) | 0.033 |
| | Block | 0.056 (0.457) | 0.020 (0.193) | 0.036 (0.364) | 0.040 (0.255) | 0.028 (0.308) | 0.012 (0.207) | 0.019 (0.357) | 0.033 |
| | All | 0.033 (0.306) | 0.018 (0.158) | 0.024 (0.195) | 0.034 (0.247) | 0.022 (0.211) | 0.018 (0.334) | 0.021 (0.331) | 0.026 |

Table 5: Steering comparison using CLIP (LPIPS) for Flux and SD3 models. Each cell shows CLIP score with corresponding LPIPS in parentheses. Micro-Avg is computed across tasks weighted by sample count.

Figure 3: Qualitative examples of SAE feature based editing across different RIEBench categories and models (SD3 and FLUX).

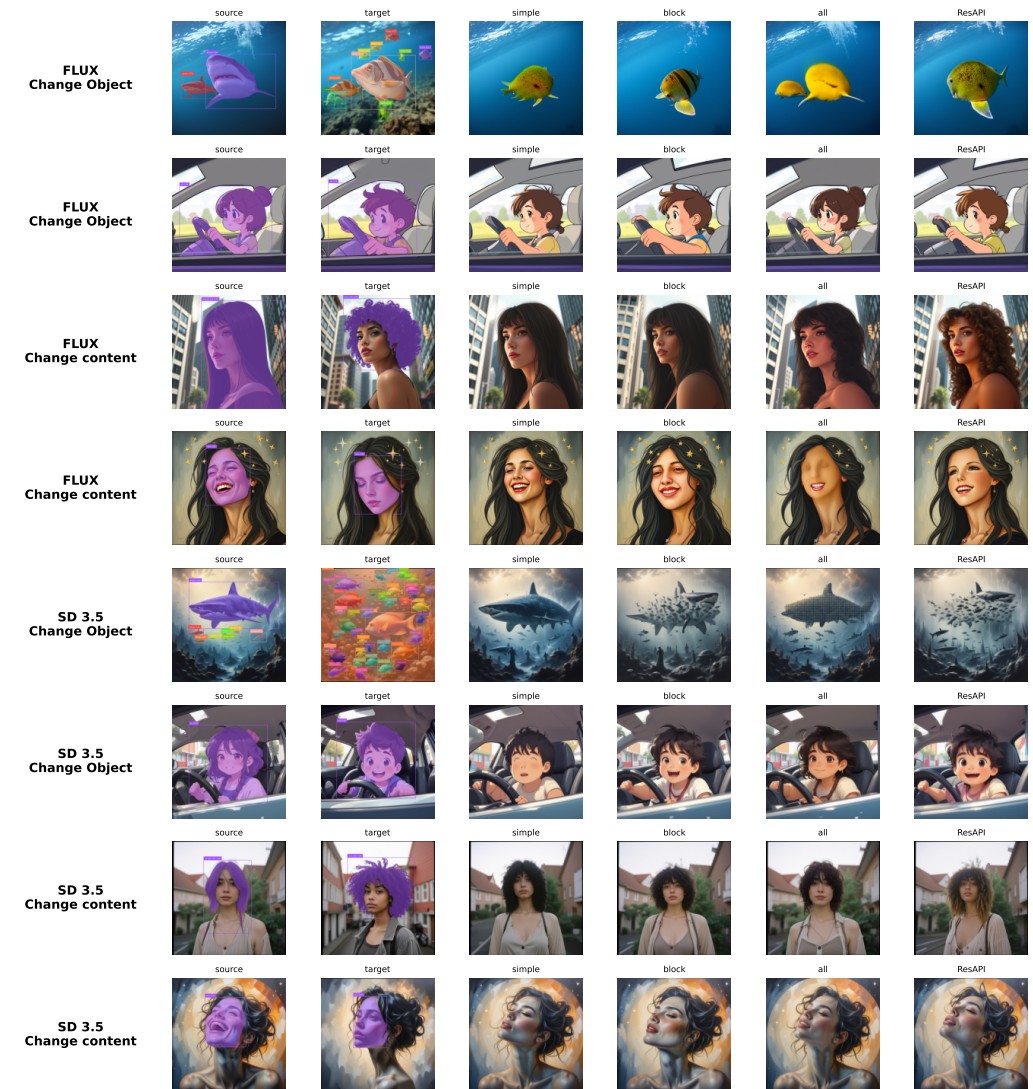

