# OpenReview forum: "Steering Diffusion Transformers with Sparse Autoencoders"
_ICLR.cc/2026/Conference — ICLR 2026 Conference Withdrawn Submission_

### Official Review · Reviewer_bEGb · 2025-10-27

**Soundness:** 2
**Presentation:** 2
**Contribution:** 2
**Rating:** 2
**Confidence:** 4

**Summary:**

In this work, the authors conduct a study of SAE-based feature steering for Diffusion transformer (DiT) models. Authors develop a similarity-based method to understand when a target concept is present in the residual stream. Using this information they propose an adaptive multi-layer steering approach which performs the injection in the layer interval where the features are naturally present. Authors demonstrate their proposed method using FLUX-Schnell and SD-3.5-Turbo on RIEBench.

**Strengths:**

* Overall I find the high level premise and the application of SAEs to steering DiT models timely and interesting.

**Weaknesses:**

* My primary concern about the work is that the experimental results are not comprehensive enough to be convincing.
	* In terms of qualitative evidence, there are only $2$ visual examples with limited coverage (cats and dogs only) in the main body of the paper. Although there is an additional figure in the Appendix, it is not clear if the adaptive multi-layer approach is any better than the "simple" baseline here.
	* In terms of quantitative results, the paper is presented as a steering method but not compared with any baselines (see [1,2] for example works). In Table $1$, $95$th percentile and maximum values of $\Delta \text{DINO}$ scores are given which is not appropriate. Reporting the average and corresponding confidence interval would be more informative about the distribution.
* Under "Contributions" (in the first page) authors mention that they introduce a framework for steering and interpreting DiTs but there is no interpretability analysis present in the paper (besides Figure 1). It would be interesting to see what concepts/features different blocks carry, how the feature lifespan are distributed for difference concepts (not only for a fixed concept and a single realization as in Figure 1).
* Please see the questions section down below as well.

**Questions:**

***Questions and Suggestions:***
* I would suggest the authors to consider rephrasing some parts of the abstract. I found some of the phrases such as "aligning interventions with the dynamics of the residual stream", and "feature's natural persistence interval" to be vague and hard to parse without reading the paper.
* I would recommend moving the SAE training details earlier of the methods section to prevent having lingering questions about "which features are used?" and "how are they collected?" to train SAE.
* I would recommend more qualitative results (visuals) in both the main body and in the appendix to make it more interesting and convincing.
* Line 422: "Diffusion transformers are often deemed “hard to steer" I am not convinced that this is universally accepted given the existing work on DiT steering (see [1,2] as examples).
* In Figure 2, images don't have too much background information but in subfigure (a) specifically, the house in the background and the lights are altered significantly when multi-steering is performed, whereas simple steering seem to preserve them better? Is subject steering and background preservation at odds to each other?
* Given that the authors present this work as a "steering" method, I would expect some comparison against established baselines and corresponding discussions on why SAE is more preferable. This is especially apparent since the analysis seem to focus mostly on the causal effects of manipulating with SAE features, but not the interpretability aspect which is lacking in the current presentation of the work.
* What was the reason for choosing to divide DiT into 10 blocks? Alternatively once you determine that Block 3 and 4 are mostly responsible for the highest contribution, did you try a more fine-grained analysis on which specific layers or even attention heads are responsible?
	* On a similar note, do you train separate SAEs for each 10 “block” or are the features combined during the training of a a single SAE?
	* Did you try focusing on specific attention heads rather than a whole transformer block as a whole?
* Did the authors observe the phenomenon of the concept appearing and disappearing in the residual stream for concepts other than cat (Figure 1)? If so, is the lifespan always always similar for all the concepts? Is there an explanation on why the concepts are consumed and vanished completely eventually?
* Why is only top $n_f$ features are kept? Based on line 198, $n_f$ is the full vocabulary size (# of concepts).
* Why does it make sense to train the SAEs on the whole updates made by a block of several layers but then later on apply the steering vectors per layer?
* Is there a metric to measure how faithful is the edited image to the context of the original image? For instance because of GroundedSAM you have access to precise locations of the background and the subject. You could measure how it is preserved after the intervention.
* How many samples are used for the numbers in Table $1$? Why are $95$th percentile and max numbers are reported here? How does the mean and median look like? It is good to report mean to understand what is the expected performance for a randomly drawn concept.
* Why does the presence of a concept in multiple layers suggest a multi layer intervention? Isn't it possible that one could do the intervention at a writer layer A and it should be present in the residual stream for however long necessary for the model?
  * Follow-up Idea: Can you adaptively understand if you need to re-inject again based on the cosine similarity? You could potentially speed-up the method by omitting to inject once you determine the information is present in the residual stream?
* Algorithm 1 in the Appendix B is difficult to follow. I would recommend the authors to make it single column.

***
***Minor Typos:***
* line 233: with -> we?
* line 397: section -> Section
* line 411: figure -> Figure
* line 416: "red" -> "blue" or "red car" -> "blue car"

***
***References:***

[1] Dalva, Yusuf, Hidir Yesiltepe, and Pinar Yanardag. "LoRAShop: Training-Free Multi-Concept Image Generation and Editing with Rectified Flow Transformers." arXiv preprint arXiv:2505.23758 (2025).

[2] Dalva, Yusuf, Kavana Venkatesh, and Pinar Yanardag. "Fluxspace: Disentangled semantic editing in rectified flow transformers." arXiv preprint arXiv:2412.09611 (2024).

---

### Official Review · Reviewer_cXY1 · 2025-10-28

**Soundness:** 3
**Presentation:** 3
**Contribution:** 3
**Rating:** 6
**Confidence:** 3

**Summary:**

This paper presents a method for steering the outputs of Diffusion Transformers (DiTs): “ResAPI”. The authors develop a multi-layer steering technique backed by feature discovery in the residual stream of these DiTs. In this work, the features in the DiT residual stream are identified using sparse autoencoders (SAEs). The authors introduce a correlation-based “concept relevance score” to identify the most concept-relevant SAE features.

**Verdict**: This paper presents a solid advancement to the mechanistic interpretability community in the realm of DiTs. The results appear promising. However, revisiting the importance of SAEs for feature selection as well as proper contextualization with other control methods for DiTs would strengthen the paper.

**Strengths:**

- S1: The paper is clearly written and the methods are well motivated and defined.
- S2: The qualitative results are impressive.
- S3: The comparisons with “simple”, “block”, and “all” are helpful for contextualizing the results of ResAPI

**Weaknesses:**

Major:
- W1: While SAEs are indeed useful for disentangling features, it seems likely that a linear probe would perform equally well or better than the SAE approach, especially since the SAE features need to be selected by associating their activations to images with similar properties (concept relevance score). The authors do state their method is “agnostic to the particular method used for feature discovery”. It would strengthen the paper to include some alternative feature selection methods.
- W2: The authors should motivate the steering approach for DiTs. While this paper does show that activation steering can be effective for DiTs, it isn’t clear that it is the optimal choice for controlled generation. Other methods such as PixArt-δ [1] also show impressive results, albeit they are not steering methods. Is steering better in terms of output generations? Speed? Efficiency? Composability? The authors should more explicitly state the benefits of activation steering in this problem setting.

Minor:
- W3: The name of the method, “ResAPI”, is unclear. Why is it introduced as an “API”? This isn’t adequately explained, and may be confusing to some readers.

[1] Chen, Junsong, et al. "Pixart-{\delta}: Fast and controllable image generation with latent consistency models." arXiv preprint arXiv:2401.05252 (2024).

**Questions:**

- Q1: How well do alternative feature selection approaches work with ResAPI?
- Q2: Why should we consider activation steering for DiTs in the first place, over other control methods?

---

### Official Review · Reviewer_7epJ · 2025-10-30

**Soundness:** 2
**Presentation:** 3
**Contribution:** 2
**Rating:** 2
**Confidence:** 4

**Summary:**

This paper studies the steering of diffusion transformers based on Sparse Autoencoder features, and introduces Residual Stream API method to detect the feature lifespan and determine the most effective steering sites. Experiments are conducted on FLUX-Schnell and SD3.5-Large-Turbo models. The method consists of searching for SAE features that present in dataset of the positive examples for some concept and are not present in the dataset with negative examples, and choosing the most relevant features, which is numerically represented by Concept Relevance Score; after the search procedure, one should perform steering by adding selected features to the residual stream on the most effective sites. Authors validate their approach using RIEBench with various image editing tasks, showing that their method allow for effective targeted steering.

**Strengths:**

S1. The paper introduces interesting method targeting the image editing and controllable generation in diffusion transformers.

S2. The application of steering via SAE to diffusion transformers is novel to my knowledge, and the paper makes interesting contributions by showing the differences between Flux and Stable Diffusion models, and the idea of feature persistence tracking (executed as Residual Stream API) seem promising.

**Weaknesses:**

W1. Practical utility and feasibility of the method application is presumably limited. It requires having a contrastive dataset of examples with and without the target concept, running the Concept Relevance Score calculation to select feature for steering, and then perform a hyperparameter search for the optimal quality of steering. It might require either having a pre-defined feature set for each concept or an expensive work for dataset creation and computations for a single concept - infeasible for end-users. Currently, the paper only provides a proof of concept that such steering method works, but without proper guidance on why and how to use it in production.

W2. Evaluation of the method is underdeveloped and it is not clear if the same or better editing quality could be achieved with other methods, and why the proposed method should be used at all. Paper focuses on simple benchmark tasks where a specific attribute must be replaced, such as object or color, which correspond to replacement of the attribute in a prompt (e.g. "Cat sat on a mat" -> "Bat sat on a mat"), but not evaluated in real use-cases scenarios where we need to make small precise changes or add new information, such as working with text, gestures and poses etc. and second-order requests like "make person on a picture happier". Many such scenarions might require slightly different workflow and dataset creation procedure, and it is unknown in what situations the approach proposed in a paper would require less effort than other methods to achieve the same editing quality or better; the prompt tuning is a simplest baseline.

W3. Limited insights into the model internals beyond the conclusion that specific layers show largest intervention effect on the RIEBench. The writing and reading behaviour of the layers is stated and speculated in the Residual Stream API methodology, but not studied.

W4. Some general claims are overstated, e.g. a single paragraph at line 107 claims that edits are precise, which is not evaluated (see W2, W3); that edits are interpretable, although interpretability of steering is never utilized or evaluated except for single example in Figure 1; that this editing is not achievable through prompting, although prompting was never evaluated as a baseline (see W2) and it is presumably wrong since use-case of the proposed method closely related to small changes in the prompt (see W2 and lines 354-357); that it does not require retraining, although true, is misleading because the search for the concept is time consuming and expensive (see W1). Similar claims are made in lines 417, 428.

W5. The ablations of method components is limited to the steering site (in section 5), but the effect of steering hyperparameters, required dataset size and reliability of the steering vector are not presented.

**Questions:**

Q1. Have you tried other feature importance and concept relevance score calculation methods? Might there be other working alternative methods to choose the right feature? A natural baseline is to rely on feature interpretations [1], it would be nice to include the comparison since it is the standard for sparse autoencoders usage. Another alternative is to discover steering vector without relying on features from SAEs, directly using the model activations [2].

Q2. It is slightly hard to understand the specific protocol for steering from the description in sections 4.3, 5.1 and algorithm 1. Do you perform steering only on those layers where you select steering vector, or you can apply it somewhere else? Am I correctly understand that you perform steering on the last layer of the block? Do you trace the feature lifespan on many blocks, not only on the block it was discovered? It would be nice to include detailed guidelines in appendix B.

Q3. In 5.1 you have calculated the averaged causal effect of the blocks. Have you identified the effects of blocks and layers for tasks or for some concepts separately? If yes, did you notice any difference?

Q4. ResAPI and Block methods apparently show comparable quality (correct me if I'm wrong), why could that be?

Q5. Different layers have different properties of hidden states geometry. Have you tried more sophisticated division into blocks, e.g. by grouping layers with comparable entropy? If yes, what are the differences? If no, can you speculate on whether it would produce more reliable results?

Q6. Residual Stream API methodology is slightly questionable in two ways. First, one uses the features from one SAE trained on specific block and applies to many layers including the ones beyond that block; one must account for internal dynamics between layers and blocks since the steering effect and concept depth are layer-dependent [3, 4], and the study of this effect is not performed in a paper. Second, a cosine similarity measurement between the steering vector and the hidden state is utilized, which might be misleading since the hidden state is a sum of many features and such cosine similarity would depend on the number of features steering vector was constructed from. The effects should be isolated. Could you provide more details to resolve these concerns or speculate on them without running new experiments?

[1] Language models can explain neurons in language models, Bills et al., 2023
[2] AxBench: Steering LLMs? Even Simple Baselines Outperform Sparse Autoencoders, Wu et al., 2025
[3] Analyze Feature Flow to Enhance Interpretation and Steering in Language Models, Laptev et al., 2025
[4] Layer by Layer: Uncovering Hidden Representations in Language Models, Skean et al., 2025

---

### Official Review · Reviewer_ujH1 · 2025-11-01

**Soundness:** 3
**Presentation:** 2
**Contribution:** 2
**Rating:** 4
**Confidence:** 3

**Summary:**

The paper proposes a practical pipeline for discovering and causally manipulating human-interpretable features in diffusion transformers (DiTs) by (1) training sparse autoencoders (SAEs) on block updates, (2) ranking SAE features with a concept-relevance score and building steering vectors, (3) locating a feature’s lifespan with a Residual-Stream similarity test, and (4) applying multi-steering (injecting the feature repeatedly across its persistence interval). Experiments on distilled DiTs (FLUX-Schnell, SD3.5-Large-Turbo) on RIEBench show that residual-aware multi-steering produces stronger, more localized edits than single-layer or naive all-layer injection

**Strengths:**

S1. One of the first systematic applications of sparse autoencoders (SAEs) to diffusion transformers (DiTs), demonstrating that SAEs trained on block updates reveal meaningful, human-interpretable features.

S2. A novel, practical method for multi-layer steering that aligns interventions with a feature’s persistence in the residual stream, producing stronger and more localized edits than single-layer injection.

**Weaknesses:**

W1. Missing SAE reconstruction / explained-variance. The paper does not report SAE reconstruction error / explained variance or sparsity statistics. Reporting these is essential whenever SAEs are applied in a new domain/dataset: low reconstruction fidelity would meaningfully weaken claims about the semantic quality of learned features.

W2. Comparisons to prior steering / feature-flow work are incomplete. The manuscript claims novelty in several places, but related steering methods that touch on feature flow and multi-layer interventions should be cited and compared. Similar work was presented here [1] and here [2], but in paper reference is missing.

W3. Ablation coverage is limited and certain hyperparameter choices are not motivated. While the paper extensively sweeps $\alpha$ and $n_f$ values, it fails to identify which parameters most significantly impact edit quality; critical ablations on alternative expansion factors (beyond the single tested value of 4), Top-k activation alternatives (JumpReLU, BatchTopK), and steering span length are missing, with the unmotivated Top-k model selection weakening the methodological rigor.

W4. Thresholding approach is under-justified. Using the maximum similarity on negatives as a per-layer threshold is simple but statistically fragile (sensitive to outliers and sample size). This can either underdetect or overdetect feature presence. More principled thresholding or significance tests are needed. Needs more ablation or justification why it will work good.

[1] Analyze Feature Flow to Enhance Interpretation and Steering in Language Models, Laptev et al., 2025

[2] Aggregate and conquer: detecting and steering LLM concepts by combining nonlinear predictors over multiple layers, Beaglehole et al., 2025

**Questions:**

Q1. The statement  on line 214 needs clarification. How is $R(i,c)$ defined and computed? Does a large positive value imply statistical significance (and if so, how is that tested)?

Q2. Equation (5) constructs the concept vector $v_c$​ as a weighted sum using mean activations. What is the rationale or empirical evidence for using average activations as weights? Have you compared this scheme to alternatives (e.g., uniform weights, weights proportional to $R(i,c)$, or learned/regression-based weights)?

Q3. What is the exact number of tokens used for each SAE?

Q4. Do you apply the steering intervention only once per DiT execution or repeatedly across the denoising steps/layers? Please explain the rationale and provide any empirical or intuitive justification for this choice.

---

### Note · Authors · 2025-12-17

I have read and agree with the venue's withdrawal policy on behalf of myself and my co-authors.